# Techniques for Profiling the Cellular Immune Response and Their Implications for Interventional Oncology

**DOI:** 10.3390/cancers14153628

**Published:** 2022-07-26

**Authors:** Tushar Garg, Clifford R. Weiss, Rahul A. Sheth

**Affiliations:** 1Division of Vascular and Interventional Radiology, Russell H. Morgan Department of Radiology and Radiological Science, The Johns Hopkins University School of Medicine, Baltimore, MD 21287, USA; tgarg3@jhmi.edu (T.G.); cweiss@jhmi.edu (C.R.W.); 2Department of Interventional Radiology, The University of Texas MD Anderson Cancer Center, Houston, TX 77030, USA

**Keywords:** immune profiling, tumor microenvironment, tissue section, single-cell characterization, spatial transcriptomics, interventional oncology, tumor

## Abstract

**Simple Summary:**

The prognosis of patients with different types of cancer is usually predicted with the help of the TNM staging guidelines; however, in recent years, there has been increased interest in using the primary tumor’s immune environment and characteristics to predict patient prognosis. Interventional radiologists play an important role in the evaluation of this immune microenvironment by performing image-guided biopsies that are paramount for obtaining appropriate tissue samples. The goal of this review is to describe the different techniques that are used for immune microenvironment evaluation, analysis steps for the data collected, and its relevance to patient care, including implications for patients undergoing interventional oncology procedures.

**Abstract:**

In recent years there has been increased interest in using the immune contexture of the primary tumors to predict the patient’s prognosis. The tumor microenvironment of patients with cancers consists of different types of lymphocytes, tumor-infiltrating leukocytes, dendritic cells, and others. Different technologies can be used for the evaluation of the tumor microenvironment, all of which require a tissue or cell sample. Image-guided tissue sampling is a cornerstone in the diagnosis, stratification, and longitudinal evaluation of therapeutic efficacy for cancer patients receiving immunotherapies. Therefore, interventional radiologists (IRs) play an essential role in the evaluation of patients treated with systemically administered immunotherapies. This review provides a detailed description of different technologies used for immune assessment and analysis of the data collected from the use of these technologies. The detailed approach provided herein is intended to provide the reader with the knowledge necessary to not only interpret studies containing such data but also design and apply these tools for clinical practice and future research studies.

## 1. Introduction

The patient prognosis in various types of cancers has traditionally been predicted with the use of the TNM staging guidelines provided by the American Joint Committee on Cancer and Union for International Cancer Control [1]. However, in recent years there has been increasing interest in using the immune contexture of the primary tumors to predict the patient’s prognosis by looking at the disease-free survival (DFS) and overall survival (OS). Tumor infiltrating lymphocytes (TILs), macrophages, dendritic cells (DC), mast cells, and cells comprise the tumor immune microenvironment. The TILs consist of T lymphocytes (TCRαβ+ T cells, CD8+ TCRαβ T cells, CD4+ TCRαβ+ T cells, TCRγδ+ T cells), B lymphocytes (antigen-presenting B cells, antibody-producing B cells, regulatory B cells) and innate lymphoid cells (natural killer cells and helper-like innate lymphoid cells). A tumor’s immune profile is characterized by analyzing at various parameters such as the density, composition, location, and functional state of TILs [2,3,4,5]. TILs play an important and diverse role in the tumorigenesis of cancers. The different subsets of TILs suppress or support the growth of tumor or metastasis through direct interactions and production of soluble factors such as growth factors, cytokines, and chemokines [6]. It has been shown that high lymphocytic infiltration such as CD3+ T cells, CD8+ cytotoxic T cells, and CD45RO+ memory T cells is associated with increased DFS and OS in patients with colorectal carcinoma, melanoma, ovarian cancer, head and neck cancer, bladder carcinoma, breast cancer, liver cancer, prostate cancer, and lung cancer [2,3,7,8,9,10,11,12,13].

One example of how data collected from immune contexture has been used to derive a simple, yet powerful classification system is called the ‘Immunoscore’. The Immunoscore is calculated after the assessment of CD3+ lymphocytes and cytotoxic CD8+ cells is performed using the Immunoscore assay in the tumor center (CT) and tumor margins (IM). The Immunoscore is calculated by looking at the number of cells in CT and TM, and based on this, the patients are divided into five groups from 0 to 4. Immunoscore has been validated internationally by the Society of Immunotherapy for Cancer, which created a consortium to globally evaluate the prognostic value of the Immunoscore in patients with stage I-III colon cancer [14]. In this study, 3539 samples were processed from 2681 patients with colon cancer [15]. The Immunoscore was found to be independently associated with time to recurrence and the rate of recurrence at 5 years was found to be significantly lower in patients with a high Immunoscore on cox multivariable analysis [15]. Immunoscore helps to capture a number of different complex mechanisms that affect the tumor progression such as presence of tumor immunogenicity (dependent on microsatellite instability, mismatch repair defect), activation of oncogenic signaling (WNT/β-catenin pathway), and differences in gut microbiota [16,17,18]. As a result of capturing these different mechanisms in one score, Immunoscore was found to be a better predictor for DFS, disease-specific survival and OS when compared to the TNM staging system, and grade of differentiation [15]. In another study conducted by Wirta et al., high Immunoscore was found to predict good 5-year survival in both microsatellite-stable and microsatellite-instable colon cancer [19]. As a result of these studies, the Immunoscore was approved for clinical, diagnostic use in patients with colorectal cancer [8,20,21]. A recent review looked at all the literature regarding the use of Immunoscore in patients with different cancers (colorectal cancer, gastric cancer, non-small cell lung cancer, pancreatic cancer, bladder cancer, head and neck squamous cell cancer, liver cancer, and ovarian cancer) [22]. In this review, all patients from these studies were pooled and the OS and DFS in patients with lower Immunoscore was found to be significantly lower when compared to those with higher score [22]. Further information about the predictive capabilities of the tumor immune microenvironment and the Immunoscore are discussed in a recent excellent review [23]. Incorporation of the Immunoscore in the traditionally used classification can lead to significantly improve prognostic and predictive tool development [14,24].

Interventional radiologists (IRs) play an important role in the biomarker-based evaluation of patients on systemically administered immunotherapies. Image-guided tissue sampling is a cornerstone in the diagnosis, stratification, and longitudinal evaluation of therapeutic efficacy for cancer patients receiving immunotherapies. Furthermore, locoregional interventions performed by IRs inarguably have a profound influence on the local tumor immune microenvironment, and strategies to manipulate local tumor immunity towards an immunostimulatory state through such locoregional interventions has the potential to overcome barriers to systemically administered immunotherapies [25,26,27,28,29]. A rigorous pursuit along these avenues of research is predicated upon the thorough characterization of the immune microenvironment after locoregional therapies. This review provides a detailed description of different technologies used for immune assessment, analysis of data collected from the use of these technologies, and the relevance of results obtained to IR practice. The heuristic approach provided herein is intended to provide the reader with the knowledge necessary to not only interpret studies containing such data but also design and apply these tools for future studies.

## 2. Tissue Section-Based Immune Profiling

Technologies based on tissue sections are amongst the most common tools used for both morphologic and molecular characterization of cancers. These tools provide several unique advantages, most notably the preservation of spatial information. Furthermore, many of these assays can be performed on archival tissue, allowing for the post hoc analysis of preserved paraffin embedded samples in tissue banks. The archetype assay for tissue section-based analysis is hematoxylin and eosin (HE) staining. This stain can be used to evaluate the TILs in sections of the malignant tissue. The use of HE staining is relatively easy in the clinical practice due to its convenience and low cost. The evaluation of HE-sections is done according to the International Immuno-Oncology Biomarkers Working Group recommendations [30]. A standard slide of 4–5 µm of formalin-fixed and paraffin-embedded (FFPE) tissue is used for TIL evaluation. When evaluating the slides, we first look at the IM and CT of the tumor under low magnification and semi-quantitatively evaluate the percent of tumor stroma and tumor cell compartment area that is occupied by mononuclear cells (at intervals of 10%) at high magnification (×200–400) [30,31,32,33]. TILs are reported as the mononuclear immune cells in the tumor stroma and tumor cell compartment separately, and they are evaluated by taking an average of the density of TILs in five different fields [33]. Specific guidelines for evaluation of TILs in different cancer types is beyond the scope of this review [30,33,34].

Immunohistochemistry (IHC) (Figure 1) uses monoclonal or polyclonal antibodies to identify specific antigens in the tissue sections [35]. A standard 4–5 µm of FFPE tissue is used for IHC evaluation [33]. In evaluation of the immune microenvironment using standard IHC, the number of CD3+, CD4+, CD8+, CD45+ cells or cancer specific antigen expressing cells are counted manually or by using a detection system in the tumor stroma, cancer cell nests and the whole microscopic field at the IM of the tumor at the magnification of ×400 [33].

### 2.1. Nanostring nCounter

Nanostring nCounter is a technology that allows direct quantification of the RNA present in the sample without the need for more complicated traditional microarray techniques such as quantitative PCR (qPCR) and RNA-seq. The sensitivity of RNA levels detected with the help of Nanostring nCounter has been found to be similar to that of traditional techniques without the need for amplification or separate enzymatic processing [36,37,38]. Additionally, when the tissue sections are fixed and embedded in paraffin, the RNA present on the tissue sections usually undergoes modification and degradation which makes the gene expression studies such as qPCR and RNA-seq challenging in these samples [39]. These techniques have demonstrated significantly worse quality of results when the input material is FFPE samples [40]. The Nanostring nCounter can be used to accurately measure the RNA expression from FFPE samples. It has been used to quantify the gene signatures from multiple tissue types [36,41,42,43]. It has been shown that the freshly prepared RNA and RNA extracted from matched FFPE samples have high correlation (r = 0.9) of transcript levels without the need for RNA purification [44]. The Nanostring nCounter works by providing a unique code to each gene being assayed. It provides multiplexed measurement of gene expression which a large number of mRNA transcripts studied simultaneously. RNA can be extracted from the FFPE slides using the QIAGEN^®^ RNeasy^®^ FFPE Kit. Nanostring nCounter kits designed to work with low quality FFPE samples with as little as 1 ng RNA can allow for analysis of a large number of gene transcripts from severely degraded samples. In total, 100 nanograms of RNA from the sample is hybridized to a custom human gene CodeSet and then processed on the nCounter platform after immobilization with Capture probe. Once the hybridization of the CodeSet with the mRNA has been completed, the next step is transferring the sample to the nCounter Prep Station. At the nCounter Prep Station, the target probes are first aligned and then immobilized in the nCounter Cartridge along with reduction in the excess probes that are inadvertently transferred to the station. The data collection then takes placed in the nCounter Digital Analyzer once the cartridge is placed in it. The target molecule of interest is identified by looking for the reporter probe which has six ordered fluorescent probes. This technology has various disadvantages such as the presence of only 800-predetermined genes which are mainly identified based on publication needs, and it does not provide any spatial information about the expression of genes.

### 2.2. Multiplex Immunofluorescence

Multiplex IHC/immunofluorescence (IF) allows for visualization of multiple markers when evaluating a tissue section, whereas conventional IHC typically only evaluates one to two markers per tissue section. In multiplex IHC/IF, the target proteins are labeled using fluorochromes and then visualized using fluorescence microscopy tools to look at the distribution of these target proteins in a single tissue section (Figure 1). This further allows for evaluation of the relative spatial distribution of, for example, various immune cell types within a tumor. Other clinical applications of multiplex immunofluorescence include understanding transduction signaling pathway activity (pAKT, mTOR, MAPK, EFGR), cell cycle characterization, inflammation, understanding autoimmune disease and neurodegenerative diseases, and identifying lymph node metastasis. Two commercial technologies currently available to perform such analysis are Vectra and CODEX. Vectra is a multispectral immunofluorescence (IF) platform that allows detection of multiple proteins within the cells along with examination of the spatial arrangement of the cell phenotypes. The Vectra platform can be used to simultaneously detect up to nine different of proteins of interest on a single FFPE tissue slide with the help of Opal chemistry and spectral un-mixing. This technology provides information about biomarker expression levels; additionally, it also increases the biomarkers that can visualized at the same time. Vectra platform can automatically load and scan up to 200 slides. After the slides are loaded, they are first pre-scanned at ×4–1 × 10 magnification which is followed by high-power-field (HPF, ×20–×40) imaging of regions of interest in the multispectral mode. The use of multiplex IF images allows comprehensive understanding of complex cellular interactions that are not possible with the use of other methods [45].

Likewise, CO-Detection by inDEXing (CODEX) is a high-parameter multiplex tissue imaging platform that allows analysis of up to 60 markers in single cells on FFPE tissue sections [46,47,48]. It uses oligonucleotide-conjugated antibodies and sequential fluorescent reports for this task. CODEX helps in the generation of detailed information about the distribution of different cellular phenotypes (‘cellular neighborhoods’) in a healthy or diseased tissue while maintaining their morphological structures [46]. It provides critical biological and clinical insight about cell–cell contacts, environmental context, and tissue structure [47]. One of the most important parts of the CODEX methodology relies on the well-designed and validated antibody panel [49,50,51]. CODEX can be used to study disease and identify biomarkers that will be useful in clinical trials to understand the cell interaction on a histological level and predict clinical outcome. Both Vectra and CODEX are Clinical Laboratory Improvement Amendments (CLIA)-certified technologies; however, the results of these technologies are required to be validated in a CLIA-certified laboratory against a gold standard (such as standard IHC), which can be a significant hurdle in the use of these technologies.

### 2.3. Spatial Transcriptomics

Over the past several years, new technologies have become available that marry the high-dimensional capabilities of bulk tissue genomic sequencing tools with the preserved spatial information of tissue section-based tools. These spatial transcriptomic tools are capable of providing unprecedented detail regarding cellular RNA and protein expression without sacrificing information from where the cells are located.

#### 2.3.1. Nanostring (GeoMX)

The GeoMX Digital Spatial Profiler (DSP) is a platform that has been developed to use the barcoding technology of nCounter in order to both spatially resolve and digitally quantify the protein and mRNA expression. GeoMX DSP can be used to analyze any sample type such as core needle biopsy, FFPE tissue block, tissue microarray, slide mounted tissue, and fresh frozen tissue based on the availability of sample. This technology uses antibodies for protein detection and RNA probes for mRNA detection. These antibodies and RNA probes are linked to oligonucleotide tags [52]. The DSP allows selection of a region of interest, which is then exposed to UV light to cause delinking of the oligonucleotide tags. Selection of the region of interest is a critical step and can be facilitated with guidance from expert pathologists [53]. These oligonucleotide tags can then be quantified using a standard nCounter assay to assess the amount of protein or mRNA present in different morphological structures. As such, the GeoMX DSP provides a spatially resolved digital profile of the analyzed tissue. The GeoMX DSP uses five different profiling modalities to define the region of interest which include contour, gridded, geometric, rare cell, and segment profiling. GeoMX DSP allows for understanding the spatial distribution of proteins and RNA which can help is discovering biomarkers to predict the patient response to therapy [26,54,55,56]. Using GeoMX DSP, it is not possible to profile every cell in a tissue slice due to its limited resolution, whereas imaging-based methods can be used to obtain multiplexed information about every cell in a tissue section [57]. Additionally, previous knowledge about the genes present is required, and only a small number of mRNAs can be investigated at once [58], though this limitation is rapidly being lifted with new integrated workflows that allow analysis of the complete transcriptome within regions of interest.

#### 2.3.2. Visium (10× Genomics)

The Visium Spatial Gene Expression platform helps with gene expression profiling at a high-resolution level [59,60]. Visium allows for spatially profiling of more than 18,000 genes by assessing their RNA expression in FFPE samples with a high resolution across the entire section of the tissue. The process begins with placing the FFPE tissue section on to the Visium gene expression slide which is stained using routine stains and images for histological purposes. The Visium slide with the stained, fixed, and imaged tissue is then loaded into the slide cassette. The RNA on the capture area (6.5 × 6.5 mm) in the FFPE slide is then bound to a capture probe. Extension of probe pairs is performed to include complements of the spatial barcodes and the sequencing library is prepared. The libraries are then sequenced, and the data is visualized to see which genes are expressed in what quantity and at which location. Comparison between NanoString GeoMX DSP and 10x Genomics Visium is shown in Table 1.

#### 2.3.3. Vizgen Merscope

Multiplexed Error-Robust Fluorescence In Situ Hybridization (MERFISH) is a microscopy-based method that allows the study of hundreds to a few thousands of genes in addition to providing single-molecular analysis capability [61]. The results of MERFISH provide subcellular spatial resolutions that are fine than sequencing-based spatial approaches. Due to the ability to study a number of target genes of interest with spatial context, MERFISH can be used along with traditional single-cell analysis.

MERFISH uses single-molecular fluorescence in situ hybridization (smFISH) in combination with labeling of RNA transcriptions using optical barcoding. MERFISH helps to assess the transcriptional activity in fixed samples by providing a highly multiplexed, single-molecule readout [61]. Targeted labeling of RNA transcripts in MERFISH is performed using a preselected gene panel in which each transcript is assigned a unique barcode [62]. After the sample is hybridized with encoding probes, the barcode is then detected by multiple rounds of multichannel imaging using different subsets of readout probes which hybridize with the barcode region of the encoding probes. The fluorescent spots are then decoded into binary barcoded (1, 0). The position of these barcodes and their positions along with the staining and segmentation of the cell nucleus and boundary allow for measurement of gene expression in a single cell. This technology can be applied to study the response of the tumor to various therapies and evaluate patient prognosis [63]. MERFISH has been used to create a spatially resolved cell atlas of mouse primary cortex, to assess the cell niche architecture of fetal liver hematopoietic stem cells, and understand cell cycle dependent gene expression [64,65,66]. He et al. demonstrated the use of MERFISH technology to perform single-cell transcriptomic imaging in FFPE tissue sections from more than 10 tissue types from mouse and human donors including breast cancer, colon cancer, melanoma, lung cancer, liver cancer, ovarian cancer, prostate cancer, and uterine cancer. They were able to map all major cell types in each tumor including the subtype of immune cells and characterize their gene expression profile at single-cell resolution [67].

### 2.4. Imaging Mass Spectrometry

Imaging mass spectrometry (IMS) uses a matrix-assisted laser desorption ionization (MALDI) for molecular study of complex biological samples such as tissue sections [68]. It allows for assessment of molecular arrangements with the need for target-specific reagents and therefore, can be useful in the discovery of various known or unknown diagnostic and prognostic markers of different cancer types and help determine appropriate therapy [69,70,71]. MALDI MIS can be used for analysis of small and large molecular mass biologicals. For analysis with MALDI, the sample is first mixed or coated with an energy absorbing matrix and then it is irradiated with the help of a laser beam. Based on the ionization mode, singly protonated molecular ions (positive ionization mode) and deprotonated ions (negative ionization mode) are generated. These ions are then detected with the help of a mass analyzer. To measure the ionized analytes in MALDI, time-of-flight (TOF) analyzers are used. The ions are accelerated at fixed potentials and the ions are then separated and recorded based on their molecular mass to charge ratio [72].

Studies using MALDI IMS have been used in the past to find molecular signatures from different tumors of different types and grades of disease. It can also be used to find spatial distribution of a wide variety of biomarkers that have been shown to play an important role in cancers, including peptides, glycans, lipids, and metabolites [73,74,75,76]. This technology allows for unbiased analysis of samples and provides multiplexed spatially resolved molecular information in addition to studying the different molecular expression in anatomically normal and pathological structures [77,78,79,80,81,82,83,84,85]. MALDI-MSI based proteomic studies have also shown to be useful in prognostic evaluation and correlating the patient response to a therapeutic regimen [80,86]. For example, Kriegsmann et al. used MALDI-IMS to classify patients with non-small cell lung cancers into squamous cell carcinoma and adenocarcinoma. A model was created using 339 different molecular signals, which was able to classify patients with 100% diagnostic accuracy. Additionally, four molecular signals (one for adenocarcinoma and three for squamous cell carcinoma) were also identified that were strongly expressed in these tissues [87]. Paulla et al. used MALDI-IMS to profile prostate cancer and benign tissues. They found a biliverdin reductase B, a biomarker which was overexpressed in the cancerous tissue [88].

Even though MALDI has been used to investigate multiple disease states, full integration of this technology into cancer research is currently not possible as proteins which are not abundantly present in the sample cannot be detected preferentially due to the lack of techniques to amplify protein signal [89].

## 3. Bulk Tissue Analysis

Analysis of disaggregated tumor tissue allows for rapid and high dimensional analysis of molecular markers within the tissue. While spatial and individual cellular information is lost, these approaches can nonetheless provide a wealth of information regarding the tumor immune microenvironment.

### 3.1. Conventional Assays

Polymerase chain reaction (PCR) is a relatively commonly used test which is used to amplify and detect the DNA and RNA sequences. In standard PCR, a set number of DNA sequences are chosen which can then be amplified to produce millions of copies for detection and analysis. Reverse transcriptase PCR (RT-PCR) is another technique that converts RNA templates in complementary DNA strands for molecular analysis. PCR is used to identify microsatellite instability in solid tumors, and metastasis workup in non-small cell lung cancers, colon cancers, and melanomas [90,91,92].

Western blots provide quantitative or semi-quantitative data about target proteins in a sample by identifying the normal proteins and modified proteins using a multi-step process [93]. Over the past couple of decades, several improvements in the preparative phase, blotting technique, and detection have occurred, which has led to the development of other methods for Western blotting. These methods include diffusion blotting, single-cell resolution Western blot, far-Western blotting, automated microfluid Western blotting, Western blotting using capillary electrophoresis and microchip electrophoresis [94,95,96,97,98,99,100,101]. These advanced techniques have been used to study drug response in tumor cells, analyze intra- and inter-tumoral variability and cell signaling [102,103].

Enzyme immunoassay (EIA) and enzyme-linked immunosorbent assay (ELISA) are used to identify the presence and concentration of different molecules in a sample. EIA and ELISA detect antigen and antibody reaction with the help of color changes by using an enzyme-linked conjugate and enzyme substrate [104]. Depending on the enzymatic activity, ELISA can be divided into homogeneous or heterogeneous which can further be divided into direct, indirect, sandwich, and competitive. The most common use of ELISA in cancer research is for quantifying the cytokine levels in biological samples. However, ELISA can measure only one cytokine in a single experiment, which makes it a time-consuming assay and impractical assay to quantify a large panel of proteins. EIA and ELISA can also be used in clinical practice to measure tumor markers such as alpha-fetoprotein, carcinoembryonic antigen, and prostate specific antigen.

### 3.2. Cytokine Analysis with Luminex

Different cell mediators such as cytokines, chemokines, and growth factors play an important role in cell proliferation, cell migration, and immune response [105]. In patients with cancers, the cytokines participate in almost all stages of tumor development [106,107,108,109,110,111]. Due to significant improvements in the cytokine detection techniques over the last 10 years, molecules in concentration as low as pg/mL can be detected from samples [112]. Most of these methods are advanced variations of the basic sandwich ELISA principle using in planar array or microbead assay multiplexed format [113]. Luminex is one such assay that allows fast and accurate measurements of cytokines by using a large number (hundreds) of specially prepared micrometer-scale plastic beads that are dyed internally using a graded mixture of red or infrared fluorescent dyes. The different degree of internal dye use creates hundreds of different fluorescent profiles which can be individually investigated and classified in a single sample [114]. A laser is then used to excite each of the unique microspheres using modified flow cytometry-based instruments and this causes them to emit different wavelengths of light. The use of Luminex is not limited to only quantification of cytokine expression patterns, it can also be used for protein expression profiling and gene expression profiling. Espinoza et al. used Luminex for cytokine profiling of the tumor interstitial fluid of the breast and compared it with normal interstitial fluid samples and matched serum samples [115]. They found increased elevation of 11 cytokines in the tumor interstitial fluid when compared to the normal interstitial fluid samples [115]. The results of this study provided information to distinguish a local tumor response in patients with breast cancer from systemic cytokine reactions [115]. Zeh et al. used ProteinChip arrays and Luminex to identify serum-based biomarkers in patients with pancreatic cancer [116]. Using ProteinChip arrays they were able to classify training data with 100% accuracy and test data with 72–83% sensitivity and 100% specificity [116]. In comparison, Luminex was able to correctly classify cases and controls with 87% accuracy with a sensitivity of 93% and specificity of 77% [116]. The results of this study show that both the technologies can be used for rapid diagnosis of patients with pancreatic cancer [116].

LEGENDplex is another bead-based assay that can be used for the measurement of multiple soluble factors such as cytokines and chemokines in serum, plasma, and cell culture supernatant [117]. Therefore, it allows for the understanding of the underlying mechanism of the disease. LEGENDplex has a distinct advantage over other based assays as it is compatible with a standard flow cytometer. It provides efficient and more sensitive multiplexing compared to traditional ELISAs. Mlynska et al. used LEGENDplex for the chemokine profiling of serum from patients with ovarian cancers and found that the presence of elevated circulating levels of C-X-C motif chemokine ligand (CXCL) 9 and CXCL10 can be used to identify immune-infiltrated tumors that would lead to shorter recurrence-free survival [118].

### 3.3. Genomic Analysis Tools

RNA sequencing (RNA-seq) allows for the study of the transcriptome of a tumor and its microenvironment and therefore, it has become an integral tool in immunogenomics. Investigators can use RNA-seq to identify tens of thousands of genes that are present in a single tumor-derived sample. RNA-seq allows the investigators to reconstruct the cellular heterogeneity of the tumor microenvironment from archival tissues [119]. It can also be used with different pathological techniques to identify different cancer subtypes, to predict response to treatment options, evaluate the patient prognosis and characterize the antigen spectrum of the tumor [120,121,122,123,124]. The evaluation of the tumor microenvironment with the help of RNA-seq provides much more data than the conventional immunological techniques such as flow cytometry or other assays that cannot use FFPE. Due to the production of a large amount of data when RNA-seq is used, it is important to have a consistent and automated workflow to prevent errors and maintain consistency. The steps include the collection of the tissue samples and isolation of RNA from it, sequencing the RNA and preparation of a library, processing of the data collected and checking its quality, and finally alignment of the data followed by final data analysis. After sequencing of the data, many different analyses can be applied to study the different gene expression, immune gene signatures, gene pathways, and T/B-cell receptor inference. The use of RNA-seq allows for greater understanding of the tumor microenvironment and helps to deconstruct it at a cellular level. Wang et al. used RNA-seq to analyze the samples of patients with primary gliomas [125]. They found that the tumor mutational burden was associated with poor outcomes in patients with diffuse gliomas such as glioblastomas [125]. RNA-seq was used by Seo et al. to identify MET exon 14 mutation has a new potential therapeutic target in patients with lung adenocarcinoma and by Nakagawa et al. to identify isocitrate dehydrogenase 1 (IDH1) as a new target in patients with chondrosarcoma [126,127]. IDH mutation has also been identified to be a good prognostic marker in patients with gliomas based on RNA-seq results by Unruh et al. [128]. RNA sequencing can also be used to identify mechanisms of cancer drug resistance as shown in a study by Shao et al. where two circRNAs were identified to be novel biomarkers and also potential therapeutic targets in patients with gemcitabine-resistant pancreatic cancer [129]. Advantages of RNA-seq include the absence of reliance on previous sequencing information, the presence of high dynamic range, the use of direct sequence alignment, and utility in identifying single nucleotide polymorphisms. Disadvantages of using RNA-seq include the high cost of processing, the requirement for high power computing facilities, the need for complex analysis of sequencing variants, and the absence of full optimization of RNA-seq protocols.

Whole exome sequencing (WES) is a sequencing technique in which the actionable areas of the genome are sequenced to identify variations in the exon regions along with causal variants of disease and/or disease-causing mutations [130,131,132,133,134,135,136]. There are currently two major approaches that can be used when performing WES; these include the short read and long read sequencing approach. Short read sequencing approach (Ilumina HiSeq X, San Diego, CA, USA) provide lower cost, more accurate data which is geared towards population level studies and for discovery of clinical variants, whereas, long read approaches (PacBio single molecule real time sequencing, Menlo Park, CA, USA) allow for de novo genome assembly applications or isoform discovery [137]. WES can be used to identify patient populations that would respond to immunotherapeutic agents such as immune checkpoint inhibitors and immunomodulatory agents in patients with solid tumors such as renal-cell carcinomas, melanomas, and non-small cell lung cancers [138]. The treatment is predicted by looking at the number of mutations present in the coding sequence of the tumor genome (tumor mutational burden) [139]. Prediction of cancers to immunotherapies such as pembrolizumab which are used for MSI-high or mismatch repair-deficient solid tumors can be assessed by identifying the MSI burden of tumors [140]. WES can also be used to identify mutational signatures of different cancer sub-types by finding different changes in the genome such as base substitutions, insertions and deletions, copy number variant changes, and rearrangements in the genome [141]. Tumor heterogeneity can be assessed by WES by quantifying the Shannon’s diversity index of the estimated single nucleotide variants [142]. Disadvantages of WES include its inability to capture all the exons in the genome, low sensitivity in the detection of structural variants and absence of sequencing of non-coding intronic regions [143,144].

Each T cell has unique antigen receptor (TCR) which are composed of alpha and beta chains whose specificity is based on the gene rearrangement process that occurs in the thymus. T cells have been shown to elicit response against its corresponding antigens (which can be cancer antigens); however, they have been shown to be driven by a wide number of T cell clones [145,146,147,148]. Due to the recent advances in next-generation sequencing (NGS) technology, TCR sequencing (TCRseq) can be carried out which allows for identification of the various TCR sequences in individuals [149]. In TCRseq, the TCR gene is first amplified and then sequenced using NGS. Based on the use of genomic DNA (gDNA based) or messenger RNA (mRNA based), the TCRseq can be divided into different methods [150,151,152]. Once the TCR sequences have been identified with the help of NGS, they are assembled into the T cell clones which share the same CDR3 sequences and are aligned to the reference sequences. The analysis of TCR sequence data can be conducted by using algorithms such as RTCT, IMGT, MixCR, and IgBLAST. [153,154,155]. TCRseq can be used to identify the developmental relationship in T cell subsets, trajectories of TILs, and identification of TCRs that respond to tumor antigens [156,157]. TCRseq can also be used to identify relative clones as shown in clinical studies where patients on anti-PD-1 antibody therapy for melanoma who responded to the treatment demonstrated clonality of the TCR repertoire in the tumor when compared to the patients who did not respond [158,159,160,161].

## 4. Single-Cell Characterization

Characterization of single-cell suspensions of immune cells has been a mainstay of immune profiling for decades. The most common approach is to use flow cytometry, and with current technologies, it is now routinely possible to stain cells with numerous markers for more detailed characterization. Multi-color flow cytometry has been developed due to the advancement in the flow cytometry hardware, software, and reagents. It provides more measurements (up to 60 parameters) compared to the conventional flow cytometry techniques and due to the more detailed descriptions of cell phenotypes, it is possible to identify rare cell populations and new cell phenotypes. It is widely used in the phenotyping of tumor cells and assessment of tumor response to various treatment modalities such as immunotherapies [162,163,164,165]. It is important to number that single-cell characterization does not provide any spatial information.

### 4.1. Mass Cytometry

Mass cytometry, which is also called CyTOF (cytometry by time-of-flight), blends flow cytometry with mass spectrometry using metal-conjugated antibodies to boost the number of detectable markers [166]. CyTOF helps to reveal the expression of targeted protein levels of each cell. The use of CyTOF can be used to identify more than 40+ parameters simultaneously and it has now emerged as a technique for large-scale immune profiling and biomarker discovery [157,167,168,169,170,171,172]. This technique allows for measurement of more markers in a single tube and therefore, fewer cells need to be analyzed per experiment compared to the traditional flow cytometry analysis to cover the similar number of markers. The CyTOF technique has been used to discover novel immune populations in humans and rodents [173,174,175]. In the CyTOF technique, the sample is first extracted for the tissue to be evaluated and an input of approximately 1,000,000 to 3,000,000 cells is required. The tissue is dissociated to achieve a single-cell suspension using the Percoll gradient. In the single-cell suspension, the cells are stained using metal-conjugated antibodies and intercalators. The samples are then analyzed using a mass cytometer and data analysis may be performed on dedicated analysis platforms such as Cytobank with SPADE, Pathsetter, and R packages [176]. CyTOF has been used in the past for the assessment of biological systems to understand response to therapy and identify signatures of disease [157,167,171,172,177,178,179,180,181,182,183]. Some of the disadvantages of using CyTOF are that it can be used only for pre-determined markers, a large number of input cells are required which makes the use of CyTOF challenging when small biopsy samples are available, cells after processing cannot be used for CyTOF, and it is affected by the sensitivity of ion used in the staining. Daud et al. used multiparametric cytometry to identify biomarkers that can predict response of melanoma patients being treated with ICI [184]. They found that high levels of CD8+, PD-1++, CTLA-4++ TILs were correlated with response to therapy and progression-free survival [184]. Additionally, they performed assessment of metastatic lesions during anti-PD-1 therapy and found a release of T cell exhaustion, which was identified by the accumulation of highly activated CD8+ T cells within tumors [184]. Similarly, in a CyTOF and RNA-seq analysis of patients with melanoma being treated with ICI the CD8+ T cell population was found to expand in tumors that displayed a CD45R0+, PD-1+, TBET+, EOMES+ phenotype [185]. Additionally, CTLA-4 blockade was found to induce expansion of ICOS+ Th-1 such as CD4+ T cells [185]. Sharma et al. used IHC and CyTOF to tumor the response of ipilimumab and tremelimumab in patients with melanoma, prostate cancer and bladder cancer [186]. They found that in patients treated with these immunotherapeutic agents there is increase in the filtration of CD4+ and CD8+ cells without change or depletion of the FOXP3 cells within the tumor microenvironment [186].

### 4.2. Single-Cell RNA Expression and Chromatin Availability Assays

Single-cell RNA sequencing (scRNA-seq) allows the study of gene expression profile of individual cells when compared to bulk RNA sequencing (RNA-seq) which provides an average of the expression profile of an heterogenous population of cells (Figure 2). The use of scRNA-seq has led to discovery of new cellular subsets in the tumor microenvironment and identify the heterogeneity between intertumoral and intratumoral cells [187,188,189]. scRNA-seq techniques follow a basic methodology in which the first step is isolation and lysis of cells, followed by generation of cDNA through reverse transcription, PCR amplification and then detection of this cDNA on next-generation sequencing platforms [190,191]. When scRNA-seq is used to analyze a collection of heterogenous cells, it can be challenging to precisely label each assayed cell. To label these cells, gene panels have been designed which can help to effectively and precisely distinguish cell labels [192]. The first scRNA-seq technique applied to primary tumor cells in order to study the different aspects of immune heterogeneity and immune response was Smart sequencing (Smart-seq) [189]. Smart-seq used isolation of single viable cells through manual pipette picking and modified dilution methods to improve transcriptome coverage [193]. The isolation of single cells can be further improved with the use of microfluidic platforms such as that of Fluidigm C1 which provides precise fluid control and broad input cell range [194,195,196]. Specific mRNA’s can be identified with the use of unique molecular identifier sequences in the generation of cDNA with techniques such as CEL-Seq (Cell Expression by Linear amplification and sequencing) [61,197,198]. scRNA-seq now allows rapid identification of markers that can affect tumor progression and change the outcomes of patients undergoing cancer treatment. scRNA-seq has been used in cancer research for understanding the response of different cancer cells to cancer therapies and for the development of resistance to different treatment options by identifying markers of resistance. For example, scRNA-seq was used to assess the triple-negative breast cancer patient derived cells that respond to epidermal growth factor receptor (EGFR) therapy and found an increased presence of enhanced stem-cell-like/mesenchymal characteristics in these cells [199]. The results of this study improved our understanding of treatment response in triple-negative breast cancer patients [199]. Jerby-Arnon et al. performed scRNA-seq of 33 melanoma tumors and identified cold niches in situ which are expressed by malignancy cells associated with T cell exclusion and immune evasion, which can predict clinical response to therapy [200]. They also found that CDK4/6 inhibition can induce repression of resistance programming in individual malignant cells leading to induction of senescence and reduce melanoma tumor growth in in vivo mouse models when given in combination with immunotherapy [200]. Sade-Feldman et al. profiled 16,291 individual immune cells from 48 tumor samples of melanoma patients treated with checkpoint inhibitors using scRNA-seq and ATAC-seq [201]. They found two distinct states of CD8+ cells, which were associated with tumor regression or progression [201]. Additionally, they found TCF7 could be visualized within CD8+ T cells and was associated with positive clinical outcomes [201].

Cellular Indexing of Transcriptomes and Epitopes by sequencing (CITE-Seq) is a technique that can be used with scRNA-seq for the detection of proteins. CITE-Seq allows uses antibodies that are conjugated to a DNA sequence that contains a PCR handle, an antibody barcode and a poly(dA) sequence [202,203]. Extension of the antibody-specific DNA sequences along with cDNAs occurs in the same poly(dT)-primed reaction due to the presence of the poly(dA) sequence. This helps in generation of a protein readout which can be captured and sequenced along with the transcriptome of a cell.

**Figure 2 cancers-14-03628-f002:**
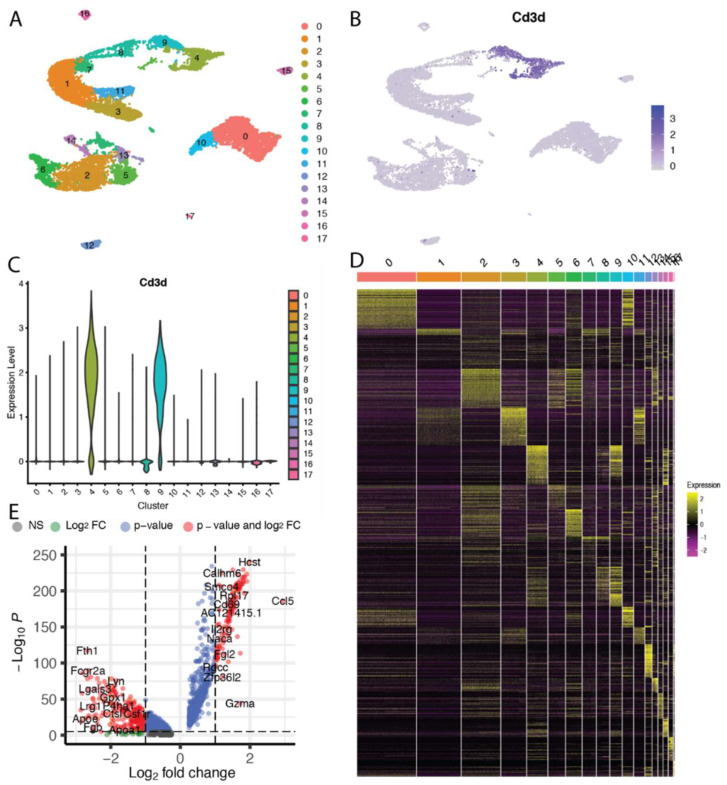
Common visualization approaches for scRNAseq data. In this example, intratumoral CD45+ immune cells isolated from orthotopic liver tumors in a rat model of hepatocellular carcinoma were sequenced using the 10X Genomics platform. Visualizations were created using the Seurat [204] and EnhancedVolcano [205] packages in R (R Foundation). (**A**) A common ‘first’ step in analyzing scRNAseq data is to discretize the cells in either a supervised or unsupervised manner into clusters of cells with similar expression patterns. These clusters can then be visualized in a t-distributed stochastic neighbor embedding (t-SNE) map where the relative similarity in expression patterns between cells is depicted in their relative proximity on a 2- or 3-dimensional diagram. (**B**) Feature maps allow for the visualization of specific genes or gene sets across the t-SNE map. In this case, the expression of the Cd3d gene was closely correlated with clusters 4 and 9, indicating that these clusters are comprised of T lymphocytes. (**C**) Another method to readily visualize the expression of genes or gene sets across clusters is with violin plots. (**D**) Heat maps are a common method to visualize differentially expressed genes across a group of samples or cell clusters. (**E**) The statistical significance as well as the magnitude of differentially expressed genes can be visualized simultaneously on volcano plots.

Assay for transposase-accessible chromatin using sequencing (ATAC-seq) is a method that can be used to explore chromatin accessibility by using a Tn5 transposase to integrates its adaptor load onto an accessible chromatin region [206]. It requires a small number of cells and can reveal the interactions between the chromatin genome location, DNA binding proteins and the transcription binding site [207]. ATAC-seq can be combined with scRNA-seq to identify extrachromosomal circular DNA in different cancer cells which are helpful in identifying the functioning of the cancer cells [208]. Additionally, it can also be combined with CITE-seq to obtain information about the surface proteins. Satpathy et al. conducted scATAC-seq to obtain chromatin profiles of more than 200,000 single cells in human blood and basal cell carcinoma [209]. They found that serial biopsies before and after PD-1 blockade can identify chromatin regulators of subsets of T cells which are therapy responsive [209]. Additionally, the study found a share mechanism that governs the intratumor exhaustion of CD8+ T cells and promotes the cell development of CD4+ T follicular helper cells [209].

Various single-cell epigenetics techniques are available that can used to understand gene regulation and cell identity by looking at cytosine modification (Bisulphite-seq), protein-DNA interaction (ChIP-seq, DamID), chromatin structure (DNase-seq), and three dimensional organization (HiC) [206,210,211,212,213,214,215,216]. The use of these single-cell epigenetic techniques will help increase our understanding of epigenetic regulation, developmental processes, and help us assess the complexity of cancers [217].

## 5. Approaches to Data Analysis

As immune profiling tools increase in depth and complexity, the amount of data generated by each assay has increased many-fold. Fortunately, the tools and workflows for analyzing these data have kept abreast of the technologies that generate the data, and numerous software packages, both commercial and open source, are available to facilitate data analysis.

### 5.1. Analysis of Multi-Color Flow Cytometry Data

Flow cytometry can help quantify a large number of parameters for millions of cells per sample; however, data analysis due to the high number of cells and parameters is often a bottleneck for the application of this technology. While commercial programs are available from multiple vendors, there are additionally easy-to-use packages for open-source platforms such as R (The R Foundation). The analysis of flow cytometry data can be categorized into six major stages: data pre-processing, cell population identification, cross-sample identification, cross-sample comparison, features extraction, and interpretation and visualization (Figure 3) [218].

#### 5.1.1. Data Pre-Processing

The flowCore package allows the users to organize and manipulate the data obtained by flow cytometry. It provides the users with an infrastructure for R-based flow cytometry analysis by organizing the data into flowFrames and flowSets. Each flowFrame reports that data along with the relative meta-data from a single experiment and multiple flowFrames make up a flowSet [219].

The flowUtils package is used along with the flowCore package. It is designed to read the Gating Markup Language files (Gating-ML) which describe the gates transferable between different software packages. The Gating ML files provides data that are computationally reproductible [220].

In the normalization stage of pre-processing step effects that arise from technical effects between samples are removed using the flowStats package. The gaussNorm and fdaNorm functions in the flowStats package normalize the data samples by identifying and aligning the high-density regions for each channel [221,222].

The quality assessment step is used to detect if there are any intersample measurement variation and it helps to identify the cause of these variations. Moreover, when the causes of variations are identified they can be removed from the analysis [218,223]. The QUALIFIER package along with flowAI and flowClean can be used to detect variations in different properties of the sample and identify them [224,225].

#### 5.1.2. Cell Population Identification

The process of identifying homogenous cell populations in the data is known as gating. This is one of the most time-consuming processes in the data analysis if carried out manually. In this process, different cell populations within a sample are identified and then compared across the sample [218]. Gating methods can be divided into sequential manual gating and automated gating.

Sequential manual gating is the traditionally used process in which the analysis is performed manually and it is based on the visual comparison of one- or two-dimensional plots [226,227]. Therefore, this step is dependent more on the expertise of the operator rather than standardized statistical application. Due to the various limitations of sequential manual gating, many automated gating methods have been developed [226,228].

Automated gating methods are based on mathematical modeling of the fluorescence intensity distribution of the cell populations [229]. Automated gating can be performed using supervised or unsupervised approaches [230]. In supervised cell population identification, a labeled dataset is required which is used to train the algorithm to learn relationships between explanatory variables and a dependent variable. flowDensity or Opencyto packages can be used for supervised automated gating [231,232]. In unsupervised approaches, a labeled dataset is not required and any predefined class can be used as a reference [218]. The data in unsupervised models are the analyses using clustering [218]. Some unsupervised gating tools include flowMeans, SPADE, and Citrus [228,233,234].

#### 5.1.3. Cross-Sample Comparison

Once the cell populations have been identified in the sample, the next step is to match the cell populations across samples to compare the characteristics of cell populations across the sample [218,235]. In this step, the characteristics of the cell population are detected, which is called feature extraction [218]. Once the features have been extracted the differences between populations of different samples are compared using meta-clustering and template construction. flowMatch and Flowmap-RF are tools which can be used for performing this step [235].

#### 5.1.4. Interpretation

In this step, different methods are used to determine associations between flow cytometry samples and class of interest such as healthy vs. unhealthy samples [236]. Supervised or unsupervised learning techniques can be used for this step. flowType, MetaCyto, and CytoCompare are three tools which can be used based on the goal of the analysis [237,238].

#### 5.1.5. Visualization

In order to analyze high-dimensional datasets, visualization of the data can be a helpful tool. With the help of visualization, different plots can be created to compare the distribution between different samples or cell populations. The flowViz package is commonly used for visualization which uses the data structures defined in the flowCore package [239].

### 5.2. Analysis of Bulk RNA-Seq Data

Genomic sequencing tools can generate hundreds of gigabytes of data per sample, and so the manipulation, let alone the investigation, of these data sets can appear daunting. However, as with flow cytometry, there are now freely available packages through platforms such as R that facilitate each step in the analysis pathway, namely, quality control, transcript identification, obtaining metrics for gene and transcript expression, and approaches for detecting differential gene expression.

#### 5.2.1. Quality Control

Quality control is needed at different steps in RNA-seq data acquisition, which includes obtaining raw reads, read alignment, and quantification in order to monitor the quality of the data. The quality check of raw reads helps to find presence of sequencing errors, PCR artifacts, or contamination. During the quality check, we look for the quality control of the raw reads, analysis of the quality of sequence, GC content, presence or absence of adaptors, and duplicated reads. FastQC and NGSQC are tools that can be used to perform the quality control on RNA-seq during raw reads [240]. These tools are similar in many aspects; however, FastQC compared to NGSQC does not allow for primer/adaptor removal, homopolymer trimming, paired-end data integrity check, quality control of 454 paired-end reads, file format conversion, and exporting high-quality or filtered reads [241]. Complete comparison of different quality control methods can be found in a recent review by Patel et al. [241]. During the read alignment step, it is important to look at the percentage of mapped reads, which is an important mapping quality parameter [242]. Other mapping quality parameters include uniformity of read coverage on exons and mapped strands. The quality control during read alignment can be performed using tools such as Picard, RSeQC, Qualimap, Bowtie2, HiSat2, RUM, STAR, and Tophat2 [242,243,244,245,246]. In a recent benchmarking study, STAR was found to have the highest number of concordant alignments, followed by Tophat2. In contrast, when looking at unmapped reads, HiSat2 and STAR outperformed the other aligners with HiSat2, producing only 1.6% of the unmapped reads [247]. Finally, after the transcript using RNA-seq has been quantified it should be checked for the GC content and gene length biases so that they can be corrected. Different R packages such as NOISeq and EDASeq can be used for this purpose [248,249].

#### 5.2.2. Transcript Identification

As the reference genome is available during the analysis of patient samples, RNA-seq analysis involves mapping of the reads onto the reference genome or transcriptome in order to identify which transcripts are expressed. After this, many of the reads may uniquely map or could be multi-mapped reads. Identification of novel transcripts using short reads is a challenging task in RNA-seq, but some methods such as Cufflinks, iReckon, SLIDE, StringTie or gene-finding tools such as Augustus can be used to increase chances of identification [250,251,252,253,254]. Similar gene finding tools have been shown to better annotate the protein coding transcripts, but their performance is worse when used for non-coding transcripts [247,255]. Of note, identification of novel transcripts on short reads is still difficult due to substantial disagreement between different methods and the low accuracy of transcript reconstruction [255].

#### 5.2.3. Transcript Quantification

Transcript quantification is the most important step of analysis as it helps to estimate the gene and transcript expression [256]. During transcript quantification step, we have to identify the number of reads that map to each transcript sequence. Programs such as HTSeq-count or featureCounts can be used for analyzing the raw counts [257,258]. However, raw read counts are not sufficient to compare the expression level between samples due to the feature-length and library-size effects. In order to look at the gene expression, therefore, we use RPKM (reads per kilobase of exon model per million reads) or TPM (transcripts per million). Commonly used algorithms that can be used for transcript quantification and include RNA-Seq by Expectation Maximization (RSEM), eXpress, and Sailfish and Kallisto [257,259,260]. In the benchmarking study by Corchete et al., Cufflinks, eXpress, HTSeq INTER, HTSeq UNION, RSEM, and StringTie were compared. The pipelines built using Cufflinks and RSEM were found to have the highest pipeline ranking followed by HTSeq and Stringtie, which also reached high-ranking positions but showed a bimodal distribution in their rank values. Additionally, the authors found that for the normalization methods, the Trimmed Mean of M values (TMM) method performed best when compared to Relative Log Expression (RLE), TPM, and fragments per kilobase exon model per million mapped reads (FPKM).

#### 5.2.4. Differential Gene Expression Analysis

Once the transcript has been quantified, the next step is differential gene expression analysis. In this step, the gene expression values are compared among samples. RPKM, FPKM, and TPM are used to normalize the sequencing depth of the samples. This process is performed by using normalizing methods such as TMM, DeSeq, PoissonSeq, and UpperQuartile, which ignore the highly variable and expressed features in order to prevent skewing of the count distribution [261,262,263,264,265]. Changes in transcript length in different samples, fragment size, GC content of the genes, and positional biases in the coverage can be corrected using the EDAseq and NOIseq R packages [249,266]. After the sample-specific normalization has been performed, the data can still suffer from batch effects, which can be minimized using batch correction methods such as COMBAT and ARSyN or using an approximate experimental design [267,268,269].

Differential expression is first computed using discrete probabilistic distribution methods such as Poisson or negative binomial [264,270]. The popular tool, edgeR, is another method, which can be used to introduce possible bias into the statistical model of the input raw reads, in order to perform a differential expression analysis. NOISeq and SAMseq are non-parametric approaches that estimate the null distribution of the data in order to perform inferential analysis by making only minimal assumptions about the data [271,272]. Finally, it is important to note that the choice of method can significantly affect the results of the analysis and the method should be chosen based on the dataset [273,274,275]. Baik et al. attempted to benchmark the workflow of differential expression analysis methods using spike-in and simulated RNA-seq data by comparing the performance of 12 differential analysis methods [276]. They found that very small dispersions and proportions of differential expression genes in the spike-in data can lead to different benchmarking results [276]. The authors have suggested appropriate methods to use in each experimental condition which can be found in the full text of the publication [276]. We would like to highlight that it is important to document the complete analysis settings and repeat it with different packages in order to find the most appropriate method for analysis.

#### 5.2.5. Visualization

Visualization of the data collected from RNA-seq can be done at the level of reads or at the level of processed coverage. At the level of reads, a commonly use tool is ReadXplorer and at the level of processed coverage genome browsers such as UCSC browser, Integrative Genomics Viewer (IGV), Genoma Maps, or Savant can be used [277,278,279,280]. Additionally, RNAseqViewer is a tool which has been developed specifically multiple RNA-seq samples [281]. Software packages such as DEXseq, CummeRbund, or Sashimi plots can be used for visualization of differential gene expression analysis results [282,283]. Due to the complexity of transcriptomes, it is difficult to efficiently display multiple layers of information, however, the currently available tools of great value in exploring results for individual genes of interest. There are multiple linear, non-linear, model-based, neural network, and ensemble method dimensional reduction algorithms that can be used to obtain low-dimensional representation of RNA-seq data. These include principal component analysis, independent component analysis, zero-inflated factor analysis, GrandPrix, t-distributed stochastic neighbor embedding (t-SNE), uniform manifold approximation and projection (UMAP), deep count autoencoder, scvis, variational autoencoder, and single-cell interpretation via multikernel learning [284]. Xiang et al. performed a comparison of these techniques using 30 simulated and five real datasets [284]. They found t-SNE to have the best overall performance with the highest accuracy and computing costs, whereas UMAP showed the highest stability with moderate accuracy and the second highest computing cost [284]. For data exploration, different graphs can be made such as density plots, histograms, boxplots, violin plots, and ridgeline plots. Density plots allow to study the distribution of a number of continuous variables. Similarly, in histograms, the continuous variables are divided into several small bins and the number of observations in each bin, and are represented by the height of each bar. Ridgeline allows for the study of the distribution of the number variables for several groups. Boxplots, and violin plots are useful for understanding the distribution of the RNA-seq data along with the expression level of different cell types. For data summary, commonly used graphs include heatmaps, dendrograms, scatter plots, and Venn diagrams. Heat maps can be used for the visualization of differential expressions of a large set of genes. They represent each individual value in a matrix as different colors based on the expression level. Dendogram is a clustering method that can be used to summarize gene expression patterns. They are designed as a network structure containing nodes and edges. Scatterplot matrices can be used to visualize read count distributions across different genes and samples and they provide a power tool for multivariate visualization. Finally, Venn diagrams can be used to compare gene expression between samples and controls and are a useful tool to compare multiple biological experiments.

#### 5.2.6. Ontology Databases (e.g., GO)

The Gene Ontology (GO) resource is a commonly used and comprehensive knowledgebase for sharing the functions of genes. The GO is structured in order to make it amenable to computational analysis, which makes it a powerful tool for comparisons and inferences about gene functions. It is structured by defining classes of gene functions that have specific relations to each other and they are often given logical definitions. Many different tools such as Categorizer, GAOTOOLS, and Map2Slim exist in order to use the GO annotations for analysis. These tools can be used to map specific GO terms to more specific GO terms [285,286,287].

Initially, the analysis of the gene expression data was performed using single-gene analysis by comparing each gene for a case and control group using parametric and non-parametric statistical tests to calculate *p*-values. After which, an adjustment to the *p*-values was made for multiple comparisons and a biological interpretation is made using these genes. However, this approach was associated with multiple disadvantages [288]. Gene set enrichment analysis (GSEA) is one approach that can be used to overcome the disadvantages of single-gene analysis by identifying enrichment or depletion levels of a set of genes of interest, called a gene set. These gene sets are selected based on the presence of a group of genes in certain biological pathways or their co-expression in different genetic conditions. A group of gene sets are then merged into collections which are called gene set databases. Some examples of gene set databases include MSigDB, GeneSigDB, and GeneSetDB [285,289,290]. More than 100 methods for GSEA have been reported in the literature; however, these methods can be categorized into over-representation analysis (ORA) methods, functional scoring (FCS) methods, and topology-based pathway (PT) analysis methods based on the differences in underlying assumptions, the notion of enrichment, null hypothesis, and significance assessment procedures [288]. ORA methods evaluate whether the fractions of genes of interest in pre-defined sets are relatively over-presented in a sub-set of the data. FCS methods aim to find changes to different elements in the dataset that can lead to alterations in a particular pathway compared to the previous knowledge. PT methods evaluate the correlations amongst genes by including the known or estimated structures of the biological networks.

## 6. Applications in Interventional Oncology

Molecular analysis of the tumor immune microenvironment starts with tissue acquisition, most typically via an image-guided biopsy performed by interventional radiology. Percutaneous needle biopsies (PNBs) are performed under imaging guidance to obtain tissue from different organs or lesions with a low risk of adverse events and producing excellent results [291]. It is important to understand the quality of tissue sample needed for different tests as the tissue requirements can vary from one molecular assay to another. A recent analysis of biopsies performed for 50 patients with hepatocellular carcinomas for genetic, proteomic, and metabolomic profiling were evaluated for the quality thresholds of the biopsy samples [292]. In total, 76.8% of the biopsies were found to be adequate for next-generation sequencing and 41.1% of the biopsies were found to be adequate for proteomic/metabolomic profiling [292]. The results of this study demonstrated the difficulty in obtaining adequate biopsies for complete profiling and the need for non-destructive point of care tests to confirm the adequacy of biopsies. Relatively tumor-rich material with a significant number of tumor cells is needed for molecular analysis [293]. Additionally, due to tumor heterogeneity, it is important to select the appropriate lesion and the appropriate component of that lesion for high-quality molecular analysis results [294]. The use of advanced imaging techniques which demonstrate the cellular components of the lesions in some cases can help select the proper site of PNBs [295].

### 6.1. Biopsy Procedure and Sample Processing

Two basic sampling techniques are used in PNBs: fine-needle aspiration (FNA) and core-needle biopsy. FNA is performed with the help of narrow-gauge needles from 20-gauge to 25-gauge, in order to obtain sufficient cells for both cytological and morphological analysis. The adequacy of samples obtained from FNA can be performed by an on-site pathologist which provides real-time feedback. Samples obtained from FNA can be used for pathological diagnosis, IHC, flow cytometry, and genetic testing. Core biopsies are performed with the help of larger needles from 16-gauge to 20-gauge, in order to obtain an appropriate amount of tissue for histological and architectural assessment of the specimen. The core biopsy samples are placed in a solution and then transferred to pathology where they are embedded in paraffin for further analysis. Selecting the appropriate type of biopsy is contingent upon a variety of factors including safety, technical feasibility, and desired molecular analysis. Excisional biopsies are the most invasive option but provide the greatest amount of tissue; furthermore, the spatial context is preserved, thus allowing for molecular analyses such as spatial transcriptomics. Accordingly, for some cancer types, excisional biopsies are the recommended approach. For example, in the National Comprehensive Cancer Network (NCCN) guidelines for B-cell lymphomas, excisional biopsies are explicitly recommended over FNA biopsies, with CNB a consideration if excisional biopsies are not feasible given the target lesion location [296]. Spatial information is preserved in CNB but given the relatively small size of the tissue sample, there is a strong potential for sampling bias; spatial analysis is not feasible with FNA samples. On the other hand, image-guided FNA and CNB are increasingly being used for molecular profiling across the cancer spectrum and even for historically ‘low-yield’ tissue such as bone metastases [291]. As minimally invasive sampling techniques improve and as tissue requirements for immune profiling tools simplify, it is likely that the role for image-guided FNA and CNB will increase. Different molecular tests and assays have different needle acquisition techniques and tissue amount requirements (number of needle passes or core samples, weight of the tissue or length of tissue core sample) which should be provided to the IRs performing the PNB’s. A heuristic understanding of the intended analysis and its attendant tissue requirements is critical for the physician performing the PNB, particularly as some molecular analyses, such as scRNA-Seq, have very rigorous tissue requirements. The tissue requirements for different tissue section-based, single-cell characterization and bulk tissue analysis techniques is provided in Table 2. In such situations, it is beneficial for the interventional radiologist to discuss with the patient’s care team how to optimize both the tissue acquisition as well as post-biopsy tissue handling and processing to maximize the chance for successful tissue analysis.

### 6.2. Evaluating the Immune Ramifications of Locoregional Therapies

The advent of immune checkpoint inhibitor (ICI)-based immunotherapies has revolutionized the landscape of systemically administered cancer therapies for solid organ malignancies. It has also stimulated tremendous excitement for the potential immunologic ramifications of locoregional therapies, including those performed by interventional radiology. In the last decade, immune profiling strategies have been used to understand the different factors associated with tumors, which can be responsible for the successful or unsuccessful treatment of cancer patients with ICI. Additionally, these technologies have been used to identify biomarkers, which can help predict the response of a particular patient to therapy [297].

There are compelling preclinical and early clinical data to suggest that interventions such as thermal ablation and transarterial embolization can augment local and systemic tumor immunity [26,27,28,298,299,300,301,302,303,304]. The thermal ablation modalities such as radiofrequency ablation and microwave ablation cause the destruction of the tumor cells by causing necrosis, whereas cryoablation cause tumor destruction via both necrosis and apoptosis. The process of cell necrosis leads to the release of cellular components into the extracellular space, which leads to an initiation of inflammatory response. Firstly, after the cell breakdown, innate immune cells react and infiltrate into the site of the tumor and later on, this is followed by activation of the adaptive immune cells [305,306]. Additionally, this process also leads to activation of an antitumor response due to the availability of tumor-specific neoantigens which now become available to the immune system. The effect of radiofrequency ablation on the immune system has been investigated in several studies. After radiofrequency ablation, heat shock protein-70 (HSP-70) has been identified as a major activator of the innate and adaptive immune systems by promoting danger signals, chaperoning antigenic peptides, and activating antigen-presenting cells in the tumor area [307]. RFA has also been shown to increase the local infiltration of dendritic cells in the tumor, enhanced CD4+ and CD8+ T cell responses, CD45RA molecule expression by naïve T lymphocytes, and CD45RO+ memory T cell increase. Increases in some of these cells, such as CD45RO+, have been found to be associated with lower rate of recurrence in patients undergoing radiofrequency ablation for the treatment of hepatocellular carcinoma [306,308,309]. The number of regulatory T cells is found to be decreased which leads to the stimulation of anti-tumor immunity and has been shown to improve survival rates [310]. Immunosuppressive cytokines such as interleukin (IL)-10 and transforming growth factor (TGF)-beta are found to be reduced whereas proinflammatory molecules such as interferon (IFN)-gamma and inflammatory cytokines such as IL-1-beta, IL-6, IL-8, and tumor necrosis factor (TNF)-alpha are found to be increased, leading to improved immune response [301,311]. Similar to radiofrequency ablation, microwave ablation also leads to release of HSP-70 after tumor destruction; however, the release of this protein has been found to be lower compared to radiofrequency ablation [312]. Microwave ablation also leads to the filtration of immune cells in the malignant tissue [313]. The circulating level of proinflammatory cytokines such as IL-1-beta and IL-6 has been found to be increased along with increases in Th1 cytokines such as IL-12 and decreases in Th2 cytokines such as IL-4 and IL-10, leading to the generation of an anti-tumor response [314]. Cryoablation due to its mechanism of cell destruction leads to both immunostimulatory and immunosuppressive effects. It has been found to increase the level of cytokines such as IL-6, IL-10, TNF-alpha, and the Th1/Th2 ratio after the procedure [315]. Additionally, the zones covered by cryoablation have been shown to have increased infiltration of CD8+, CD4+ Y cells and FOXP3+ cells [316]. There is paucity of data regarding the characterization of immune response after immunotherapy. A recent study by Tischfield et al. characterized the transarterial embolization as inducing modulation of the TME using a rat model. They observed normalization of cirrhosis induced alternation in levels of CD8+, CD4+, and CD25/CD4+ lymphocytes, and increased in number of CD3+, CD4+, and CD8+ TILs [298].

Intratumoral immunotherapies can also be used to further unlock the potential of image-guided procedures by delivering potent immunologically active therapies directly into the tumor [25,29,317,318]. To drive the field forward, a thorough appreciation for the mechanistic underpinnings of locoregional therapies and their influence on the tumor immune microenvironment is necessary.

## 7. Conclusions

This review provides a detailed description of tissue-section based immune profiling techniques, bulk tissue analysis techniques, single-cell characterization techniques, workflow for analysis for the data collected from these techniques, and a focus review of sampling techniques and specifications along with immune changes after interventional oncology procedures. It is our hope that an empiric appreciation for the state-of-the-art immune profiling tools in this review will facilitate incorporation of these technologies into ongoing and future studies on locoregional therapies so that the key questions in the field can be answered.

## Figures and Tables

**Figure 1 cancers-14-03628-f001:**
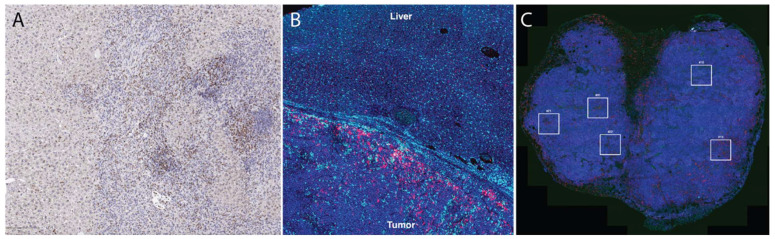
Tissue section-based immune profiling tools. (**A**), conventional immunohistochemistry of an orthotopic mouse liver tumor demonstrates infiltration of CD8+ cells (DAB, brown color stain) into the tumor. (**B**), immunofluorescence of CD68+ (light blue) and CD8+ (magenta) illustrates accumulation of macrophages and T cells at the liver-tumor border in a rat model of liver cancer. (**C**), Multi-color immunofluorescence (pan-cytokeratin = magenta, blue = DAPI, red = CD3, green = CD8) is helpful to characterize accumulation of multiple cell types; in this case, these images were used to draw regions of interest (boxes) within which high-dimensional RNA sequencing was performed using the Nanostring GeoMX platform [26].

**Figure 3 cancers-14-03628-f003:**
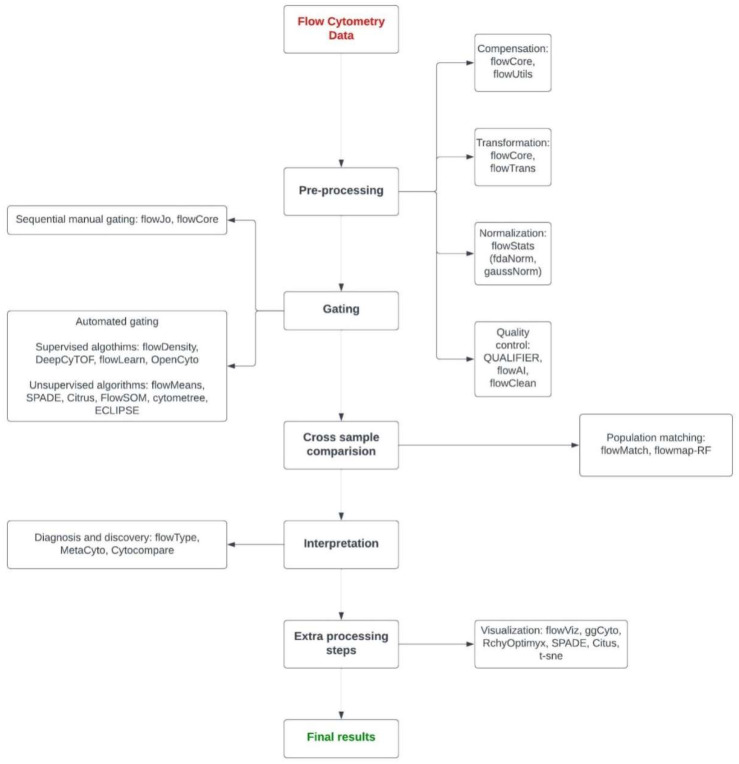
Steps for analyzing flow cytometry data along with packages that can be used at each step.

**Table 1 cancers-14-03628-t001:** Comparison between commonly used commercially available spatial transcriptional profiling platforms (Adapted from Bassiouni et al. [60]).

Comparison Parameters	10× Genomics Visium	NanoString Ge-oMX DSP
Tissue compatibility	Compatible with fresh frozen tissue	Compatible with fresh frozen tissue and FFPE
RNA quality	Needs RNA integrity number >7	No requirements for RNA quality
Tissue preparation	Tissue is mounted on a specialized gene expression slide	Tissue is mounted on a standard microscope slide
Tissue size	6.5 × 6.5 mm per capture area	14.6 × 36.2 mm
Detection area	Within the full capture area	Within user-defined regions of interest
Cellular resolution	~10 cells/feature	~20–200 cells/region of interest
Direct RNA detection	Absent	Present
Concurrent protein detection	Present	Present

**Table 2 cancers-14-03628-t002:** Tissue and preparation requirements for different techniques.

Technique	Tissue Requirements
**Tissue Section-Based Immune Profiling**	
Nanostring nCounter	FFPE-derived RNA: 300 ng Total RNA: 50–100 ng Fragmented DNA: 300 ng ChiP DNA: 10–100 ng (based on presence or absence of amplification) Single cell: 8 µL of amplified sample
Multiplex immunofluorescent (CODEX and VECTRA)	Tissue section with thickness of 5–10 µm
GeoMX Digital Spatial Profiler	Small cells: 10–20 cells for protein, 50–200 cells for RNA Large cells: 1–5 cells for protein, 5–20 cells for RNA
Visium (10× Genomics)	≤6.5 × 6.5 mm tissue section
Vizgen Merscope	Tissue block size up to 1.5 cm^3^
Matrix-assisted laser desorption ionization (MALDI)	0.05 µL of sample mixed with 0.45 µL of matrix
**Bulk Tissue Analysis**	
Polymerase chain reaction	For solid tissues 25–50 mg
Western blots	20–50 mg of cellular lysate
Enzyme immunoassay	100 µL
Luminex	25–50 µL
RNA-seq	>2 µg or >50 ng/µL
Whole exome sequencing	500 ng
TCR-seq	FFPE: 25 µm in 100 µL Sorted cells: 10–25 ng/µL or 1–3 µg
**Single-Cell Characterization**	
Cytometry by time-of-flight	1 × 10^6^ cells/mL
scRNA-seq	>1 million cells in 1 mL
CITE-seq	1–2 million cells in 0.1 mL of staining buffer
ATAC-seq	>1 million cells in 1 mL

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
