# Peer review of "Techniques for Profiling the Cellular Immune Response and Their Implications for Interventional Oncology"

_cancers, 2022, doi:10.3390/cancers14153628_

Round 1

Reviewer 1 Report

The authors have done an excellent job responding to comments.  Given the change in paper focus to predominantly immune-analysis techniques I would consider a change in title that de-emphasizes the role of interventional oncology procedures.  Consider "Techniques and Consideration for Cellular Immune Response Profiling with Implications for Interventional Oncology Procedures".

Also note the paper needs an additional read for grammatical errors.  For example, in the Simple Summary alone: line 12 - in the recent years there is interested interest in...line 16 - the different techniques that are used of immune microenvironment...

Author Response

Response to Reviewer 1

The authors have done an excellent job responding to comments.  Given the change in paper focus to predominantly immune-analysis techniques I would consider a change in title that de-emphasizes the role of interventional oncology procedures.  Consider "Techniques and Consideration for Cellular Immune Response Profiling with Implications for Interventional Oncology Procedures".

We thank the Reviewer for their helpful comment. We have revised the title to the following: “Techniques for Profiling the Cellular Immune Response and their Implications for Interventional Oncology”.

Also note the paper needs an additional read for grammatical errors.  For example, in the Simple Summary alone: line 12 - in the recent years there is interested interest in...line 16 - the different techniques that are used of immune microenvironment...

We thank the Reviewer for their helpful comment. We have edited the manuscript to correct multiple grammatical and syntax errors.

Reviewer 2 Report

The authors have done a commendable job in markedly improving the quality and structure of the review. This is now a well written review which is absolutely of use to the field.

Author Response

We thank the Reviewer for their positive comments.

This manuscript is a resubmission of an earlier submission. The following is a list of the peer review reports and author responses from that submission.

Round 1

Reviewer 1 Report

This is a nice review of immune-analysis techniques that will have interest to any translational clinical oncology researcher interested in correlative biopsy immune-analysis techniques.  

This reviewer agrees with the authors that a better understanding of the immune-effects of interventional oncology procedures is important to advance the field, but as the review currently reads it is not clear that this is an equal focus of the paper.  This concept is not discussed in detail until the Discussion/Conclusion, with only brief mention in the title and a single sentence in the Introduction. The relevant articles are not cited until the 200s.  Recommend including mention of the immunologic impact of interventional oncology procedures within Simple Summary and Abstract to provide a different frame for the review.  Also recommend devoting more time to discussing these concepts within the introduction of the paper 

Regarding the immune impact of interventional oncology procedures, certain important primary research articles are not directly cited - noted omissions include Waitz et al Cancer Research 2012, Duffy et al J Hepatology 2017, Also the high-quality review t"Oncolysis without viruses ..." by Kepp et al Nature Reviews Clinical Oncology 2020 is worthy of citation. 

Additional comments:

Figure 2 in text refers to images in second and third panel of Figure 1 (multiplex IF)

Figure 2 (sc-RNAseq graphic) is not called out in the text

"Biopsy procedure and sample processing" should be it own section, not a sub-section of Discussion/Conclusion, and should come in the text body.  It would be helpful when possible to include actual details of sample processing (for example, samples intended for sc-RNA-seq should not be placed in formalin)

Reviewer 2 Report

The authors should be congratulated on the excellent, clear writing. There are very few grammatical and syntax errors with the exception of the occasional misuse of linking words which can be easily corrected.The authors' clarity and hard work in putting together this review is, however, unfortunately diminished by a number of glaring oversights , in particular:

1. The authors premise the entire review from the perspective of post-processing of IR-gleaned samples. However, very little attention is paid to the importance of how and when such samples are biopsied, processed and analyzed. Neither is mention made of the potential pitfalls inherent in biopsies from such sights such as failure to encapsulate tumor heterogeneity in a single biopsy. No mention is made of how next generation radiological modalities can be used to inform biopsy sites (eg PET-directed) or when a patient should receive a biopsy or the use of non-biopsy approaches such at circulating tumor DNA which may better suited to summarize the heterogeneity of tumor samples. 

2. The manuscript currently reads as a repetitive list of techniques with no real critical insight into the pros, pitfalls and potential future directions of these methods. The reader would be greatly advantaged by understanding which are the "best" methods available and thus how they could integrate that into their research and practice.

3. The clinical relevance of the addressed techniques is either mentioned only in passing or completely overlooked (see attached PDF comments for specific points).

4. There are some important techniques and studies that are not mentioned. For example, differential gene expression and the growing utility of benchmarking studies to compare bioinformatic workflows (see attached PDF comments for specific points).

5. The structure is rather non sequitur. Discussions regarding sample acquisition and processing should be at the start. Discussions regarding bulk sample analysis should be before single cell analysis. 
